# Dual-Energy CT, Virtual Non-Calcium Bone Marrow Imaging of the Spine: An AI-Assisted, Volumetric Evaluation of a Reference Cohort with 500 CT Scans

**DOI:** 10.3390/diagnostics12030671

**Published:** 2022-03-09

**Authors:** Philipp Fervers, Florian Fervers, Mathilda Weisthoff, Miriam Rinneburger, David Zopfs, Robert Peter Reimer, Gregor Pahn, Jonathan Kottlors, David Maintz, Simon Lennartz, Thorsten Persigehl, Nils Große Hokamp

**Affiliations:** 1Department of Diagnostic and Interventional Radiology, Faculty of Medicine and University Hospital Cologne, University Cologne, 50937 Cologne, Germany; mathilda.weisthoff@uk-koeln.de (M.W.); miriam.rinneburger@uk-koeln.de (M.R.); david.zopfs@uk-koeln.de (D.Z.); robert.reimer@uk-koeln.de (R.P.R.); jonathan.kottlors@uk-koeln.de (J.K.); david.maintz@uk-koeln.de (D.M.); simon.lennartz@uk-koeln.de (S.L.); thorsten.persigehl@uk-koeln.de (T.P.); nils.grosse-hokamp@uk-koeln.de (N.G.H.); 2Fraunhofer Institute of Optronics, System Technologies and Image Exploitation (IOSB), 76131 Karlsruhe, Germany; florian.fervers@iosb.fraunhofer.de; 3Philips CT Clinical Science, 22335 Hamburg, Germany; gregor.pahn@philips.com

**Keywords:** bone marrow, spine, tomography, X-ray computed, artificial intelligence

## Abstract

Virtual non-calcium (VNCa) images from dual-energy computed tomography (DECT) have shown high potential to diagnose bone marrow disease of the spine, which is frequently disguised by dense trabecular bone on conventional CT. In this study, we aimed to define reference values for VNCa bone marrow images of the spine in a large-scale cohort of healthy individuals. DECT was performed after resection of a malignant skin tumor without evidence of metastatic disease. Image analysis was fully automated and did not require specific user interaction. The thoracolumbar spine was segmented by a pretrained convolutional neuronal network. Volumetric VNCa data of the spine’s bone marrow space were processed using the maximum, medium, and low calcium suppression indices. Histograms of VNCa attenuation were created for each exam and suppression setting. We included 500 exams of 168 individuals (88 female, patient age 61.0 ± 15.9). A total of 8298 vertebrae were segmented. The attenuation histograms’ overlap of two consecutive exams, as a measure for intraindividual consistency, yielded a median of 0.93 (IQR: 0.88–0.96). As our main result, we provide the age- and sex-specific bone marrow attenuation profiles of a large-scale cohort of individuals with healthy trabecular bone structure as a reference for future studies. We conclude that artificial-intelligence-supported, fully automated volumetric assessment is an intraindividually robust method to image the spine’s bone marrow using VNCa data from DECT.

## 1. Introduction

Computed tomography (CT) of the chest and abdomen is the recommended and most frequently conducted imaging procedure for the staging of malignant disease [1,2,3,4,5]. Due to rapid scanning times, good patient acceptance, and few contraindications, it is also among the most commonly performed imaging procedures in the Western world, with ever-rising numbers [6,7,8]. A well-known limitation of contrast-enhanced CT, compared with, for instance, magnetic resonance imaging (MRI), positron emission tomography CT (PET/CT), or bone scintigraphy, is its limited capacity to accurately diagnose metastatic disease of the spine. The dense trabecular structure of vertebral bodies impedes the assessment of underlying, malignant tumors without dominant osteolytic or osteoblastic components. In a recent meta-analysis, conventional CT yielded a pooled sensitivity and specificity of 0.77 and 0.83 for only the detection of spine metastasis [9]. Since the thoracolumbar spine is among the overall most common locations of bone metastases, this diagnostic gap of conventional CT represents a major clinical limitation [10,11]. Additional MRI, PET/CT, or scintigraphy significantly improves the detection of metastatic spine disease but at an additional cost of radiation exposure or extended contraindications. Moreover, additional exams delay diagnosis, increase patient discomfort, and raise economic expenses [9].

By now, dual-energy CT (DECT) is a widely available technology, which holds the potential to narrow the diagnostic uncertainty of conventional CT concerning metastatic spine disease [12,13,14]. DECT exploits the physical phenomenon that the interaction of an X-ray beam with any absorbing material depends on a material-specific combination of the photoelectric effect and Compton scattering. Approximating the contribution of either separately allows for material decomposition, e.g., for calcium [15,16]. In the context of spine imaging, the postprocessing of virtual non-calcium (VNCa) images is of particular interest. Using voxel-by-voxel material decomposition, VNCa images emulate Hounsfield units (HUs) without the calcium-specific portion of X-ray attenuation [12]. This technique aims to virtually remove the trabecular structure of the vertebral body and enables dedicated bone marrow imaging of the spine [12]. Several recent studies suggest similar capabilities of VNCa images, compared with the gold standards MRI and PET/CT, when assessing malignant infiltration of the spine’s bone marrow [12,17,18,19,20,21,22,23] or traumatic vertebral bone marrow alterations [24,25]. While study-specific cutoffs have been provided for these purposes, knowledge on the distribution and variability of VNCa values encountered in healthy individuals is sparse.

This lack of reference values for the physiological bone marrow attenuation on VNCa images obligates each comparative study to define an individual reference cohort and furthermore impairs real-world applicability of findings from the literature. Hence, the aim of our study is to provide a large-scale reference cohort of physiological bone marrow attenuation on VNCa, as a foundation to support further investigation of pathological bone marrow alterations.

## 2. Materials and Methods

All procedures performed in studies involving human participants were conducted in accordance with the ethical standards of the institutional (Application Number 21-1105) and national research committee and with the 1964 Helsinki declaration and its later amendments or comparable ethical standards. Informed consent was waived due to retrospective study characteristics.

### 2.1. Patient Enrollment

Inclusion criteria to our study comprised the following factors:(1)Intravenous contrast-enhanced DECT of the chest and abdomen by the protocol specified below;(2)Examination performed between 1 January 2016 and 1 January 2021 after resection of a malignant skin tumor;(3)Patient age >18 years.

Exclusion criterium was macroscopic evidence of metastatic disease.

Metastatic disease was excluded in consensus reading by two experienced radiologists (with 4 and 7 years of experience in oncologic imaging). A total of 500 DECT scans conforming to the enrollment criteria were randomly selected as our study population.

### 2.2. DECT Imaging Protocol

Patients were examined in a head-first, supine position on a commercially available DECT scanner (IQon Spectral CT, Philips Healthcare, Amsterdam, The Netherlands). All scans were performed after intravenous contrast administration in the portal venous phase, using bolus tracking in the descending thoracic aorta (delay of 50 s, threshold 150 HU). Scan parameters were as follows: tube voltage 120 kV; tube current modulated by DoseRight 3D-DOM (Philips Healthcare); collimation 64 × 0.625 mm; pitch 0.671; a total of 100 mL of iodinated contrast medium (Accupaque, GE Healthcare) was administered by a 20 g intravenous catheter with a flow rate of 3.0 mL/s, followed by a saline flush of 30 mL.

### 2.3. DECT Image Reconstruction and Postprocessing

All images were constructed in a 512 × 512 matrix and a slice thickness of 2 mm with an overlap of 1 mm. Spectral-based raw data were processed to VNCa images using the vendor’s proprietary software (IntelliSpace Portal, Spectral Diagnostics Suite, Philips Healthcare) employing maximum, medium, and low calcium suppression indices (index 25, 50, and 76, respectively).

### 2.4. Automated Segmentation of the Bone Marrow

Data assessment in our study was fully automated and did not require specific user interaction. First, the spine was segmented on conventional images employing a pretrained, convolutional neuronal network, which won the VerSe Vertebrae Segmentation Challenge by the Technical University of Munich in 2020 [26,27]. The neuronal network by Payer et al. ranked first, with a dice coefficient of 0.94 and a correct vertebrae labeling rate of 0.99, compared with expert readings of >300 CT examinations [28]. In the second step, excess vertebrae above a maximum number of 17, counted from the bottom one (5 lumbar and 12 thoracic vertebrae), were identified by a Python script. We discarded excess vertebrae above 17 to avoid inconsistent inclusion of the partially imaged cervical spine. Consecutively, the resulting volume of interest (VOI) containing the thoracolumbar spine was narrowed at all margins by 3 mm, using the SciPy command “scipy.ndimage.binary_erosion”. This process aimed to exclude the vertebrae’s bordering cortical bone, which does not contain bone marrow, resulting in the three-dimensionally segmented thoracolumbar spine’s bone marrow space. Lastly, the VOI mask was automatically transferred to the postprocessed VNCa images at low, medium, and maximum calcium suppression settings (Figure 1). Bone marrow attenuation histograms with a bin size of 5 HU were yielded by a Python script at each suppression level. Data visualization was achieved by 3D Slicer [29].

### 2.5. Statistical Assessment

Statistical analysis was performed in the R language for statistical computing (version 4.0.0, R Foundation, Vienna, Austria). For further analysis, bone marrow attenuation histograms were normalized to a standard volume. Attenuation values below −1000 HU in the bone marrow space were not respected, since they appear due to VNCa postprocessing of densely calcified structures (e.g., cortical bone islands, segments of cortical bone), which do not contain bone marrow [23]. Attenuation histograms were reported by patient age, sex, and the level of calcium suppression (low, medium, maximum). The intraindividual consistency of our automated method was assessed by the percentage overlap of attenuation histograms at two consecutive examinations. Similarly, the age- and sex-specific bone marrow attenuation profiles were compared by the overlap of attenuation histograms (Figure 2). Figures were plotted using the R library ggplot2 [30].

After segmentation of the thoracolumbar spine by a convolutional neuronal network, intraindividual consistency (panel A) was assessed by the overlap of normalized bone marrow attenuation histograms in consecutive exams. Patient A underwent two dual-energy CT scans within 119 days (gray and turquoise area), yielding an overlap of bone marrow attenuation of 0.95, which locates close to the median overlap in our study population. Panel B illustrates the interindividual overlap of bone marrow attenuation between males and females in the age group of 21–40 years, presented by the sex- and age-specific average attenuation histograms (overlap 0.81). It is worth noting that for reasons of simplicity, only data with maximum calcium suppression levels are illustrated.

## 3. Results

We evaluated 500 DECT scans of 168 patients (88 female, mean age 61.0 ± 15.9 years). The median number of exams per patient was 2 [2,3,4], the median time interval between two examinations was 185 [160–231] days. A total of 8298 thoracolumbar vertebrae were segmented with a median of 17 [16,17] vertebrae per exam. This corresponds to the inclusion of 97.6% of all possible thoracolumbar vertebrae (8298 segmented vertebrae/500 DECT scans × 17 thoracolumbar vertebrae). Batch processing of DECT data was not interrupted for manual alterations.

The intraindividual consistency of our method was assessed by the overlap of attenuation histograms. The median overlap of bone marrow attenuation of consecutive exams yielded an overall 92.9% [87.8–95.6]. The median intraindividual overlaps in low, medium, and maximum VNCa settings were 93.7% [90.1–96.2], 91.6% [87.1–95.4], and 92.4% [86.8–95.1], respectively. Notably, 92.8% and 65.3% of follow-up examinations achieved at least a “good” and “excellent” intraindividual consistency, respectively (overlap of >80.0% and >90.0%, respectively). The study sample and basic results are illustrated in Figure 3.

### 3.1. Quantitative Features of Physiological Bone Marrow Attenuation

Bone marrow attenuation was assessed on low, medium, as well as maximum calcium suppression settings, and grouped by patient age and sex. Table 1 reports median attenuation values along with quartiles and 95th percentiles, and the location of the histogram’s maximum for each group. Comprehensive attenuation histograms are plotted in Figure 4.

### 3.2. Age- and Sex-Specific Attenuation of Bone Marrow

Bone marrow attenuation distribution was not uniform but varied depending on patient age and sex. The largest spread of bone marrow attenuation profiles was observed in the maximum calcium suppression index (Figure 4), which is the most frequently used VNCa setting for bone marrow imaging in recent literature [12,23]. In higher calcium suppression levels, bone marrow attenuation was found generally lower in younger age. The age-adjusted, sex-related differences in bone marrow attenuation were less pronounced (Figure 5).

## 4. Discussion

DECT has recently been suggested to overcome the limited diagnostic accuracy of CT in diagnosing metastatic disease of the spine. Particularly, VNCa images postprocessed from DECT data have shown the potential to close the diagnostic gap and suggested similar capabilities to MRI and PET/CT for detection of spinal metastasis [12]. To facilitate the clinical transition of reports on VNCa performance in diagnosing occult malignant disease of the spine by conventional CT, this study sought to provide reference values for VNCa attenuation of the spine’s bone marrow among healthy individuals across different age groups and sexes, which are missing in literature to date.

In order to avoid bias, we avoided specific user interactions when identifying the reference values. Artificial-intelligence-based data processing allowed for a fully automated methodology, aiming at maximum generalizability of our results. Notably, the intraindividual overlap of attenuation profiles, as a check for consistency of our methodology, was excellent (median overlap of consecutive bone marrow attenuation histograms 92.9%). The excellent consistency in consecutive examinations suggests that VNCa bone marrow attenuation is a robust, reproducible imaging parameter. As our main result, we provide an in-depth description of the normal ranges of age- and sex-specific bone marrow attenuation profiles.

A recent study has successfully identified an attenuation cutoff on VNCa images for detection of metastatic disease of the spine but without acknowledgment of the age- and sex-specific changes in physiological bone marrow attenuation [12]. Abdullayev et al. investigated a cutoff between −174.9 HU and −143.2 HU on maximum VNCa suppression settings for diagnosis of spinal metastasis, using contrast-enhanced DECT in a relatively young patient population (mean age 56 years, 71.4% female) [12]. Using identical technology and a similar imaging protocol, our reference cohort supports their findings for the evaluated patient cohort, since both cutoffs locate above the respective age- and sex-specific 95th percentile of physiological bone marrow attenuation (age 41–60 years, female: reference 95th percentile −212.5 HU, Table 1). Malignant infiltration displaces the fatty, healthy bone marrow by soft tissue tumor, raising the bone marrow’s VNCa attenuation [17]. This hypothesis has been recently investigated for bone marrow infiltration by multiple myeloma on non-contrast-enhanced VNCa images [18,20,23]. A shift in VNCa attenuation above the 95th percentile of the here-suggested age- and sex-matched reference range in the context of malignant disease might serve as a desirable imaging biomarker in the future.

Definition of the upper margin of a normal reference range at the 95th percentile is a common practice to achieve an arbitrary distinction between a physiological vs. pathological result of a medical test [31,32,33,34]. Adjustment for basic patient demographics, such as age and sex, however, is crucial for reliable results [33,34]. Recalling the study by Abdullayev et al., the investigated cutoffs for the malignant disease were above the 95th percentile of their study sample; however, they were located in the normal range of bone marrow attenuation of elderly females (age > 80 years, female: reference 95th percentile −122.5 HU, Table 1), rendering them physiological in such patients [12]. Our data suggest a large disparity of bone marrow attenuation between younger and elderly individuals, particularly pronounced among women. Hence, without adjustment for sex and age, arbitrary cutoffs for the diagnosis of bone marrow disease are of limited use.

Traditionally, bone marrow imaging was a domain of MRI, and several investigations aimed to outline the normal ranges of physiological bone marrow appearance [35,36]. In particular, MRI is feasible to image the physiological conversion from cellular, red bone marrow to fatty, yellow marrow in healthy individuals [35]. Our data demonstrate higher bone marrow attenuation with rising age, which cannot be explained by fatty conversion since fat demonstrates lower attenuation than cell-rich tissues. Hence, we suggest that the age-dependent increase in bone marrow attenuation in our study is a result of decreasing bone mineral density, i.e., virtual calcium suppression of voxels with higher calcium-like attenuation results in relatively lower VNCa attenuation. Similarly, iodine uptake of the bone marrow might affect VNCa attenuation: Thus far, most VNCa studies of malignant bone marrow disease are restricted to non-contrast-enhanced scans [17,18,19,20,21,22,23]. A possible explanation is that virtual suppression of calcium-like attenuation might interfere with iodine-like attenuation [12,37]. In particular, when using higher VNCa settings, besides the desired suppression of calcium-like attenuation, a portion of the iodine-like attenuation is also virtually suppressed [12]. This technical aspect of VNCa postprocessing, however, is a well-known and reproducible limitation [12,38,39]. However, compared with our investigation, which aimed to examine the overlap of material-specific attenuation, a recent phantom study using an identical spectral detector CT (SDCT) scanner suggested excellent capability to separate calcium- and iodine-like attenuation [40]. Abdullayev et al. also made use of the identical SDCT scanner to successfully identify metastatic spine disease on contrast-enhanced VNCa images, which might serve as well as a practical proof of concept [12]. Several further authors consider this approach promising and warrant further investigation [13,14].

Our study has several limitations that need to be discussed. First, the values of normal ranges we report are limited to the specific scanner used; nevertheless, we consider our methodology transferable to other imaging protocols. Second, while macroscopic tumor burden was excluded based on image assessment, all patients had a history of local dermatological malignancy (which was considered successfully treated). However, microscopic tumor burden might be present at the timepoint of imaging. Last, the analyzed patient population demonstrated a wide demographic spread; however, it was not truly normally distributed; the age and sex distribution resembled patient populations commonly encountered in oncologic imaging in daily practice.

## 5. Conclusions

In conclusion, we provided the first, large-scale reference cohort of healthy individuals in VNCa bone marrow imaging from SDCT. We outlined age- and sex-specific normal ranges of bone marrow attenuation to facilitate the clinical transfer of DECT-based assessment of metastatic disease of the spine.

## Figures and Tables

**Figure 1 diagnostics-12-00671-f001:**
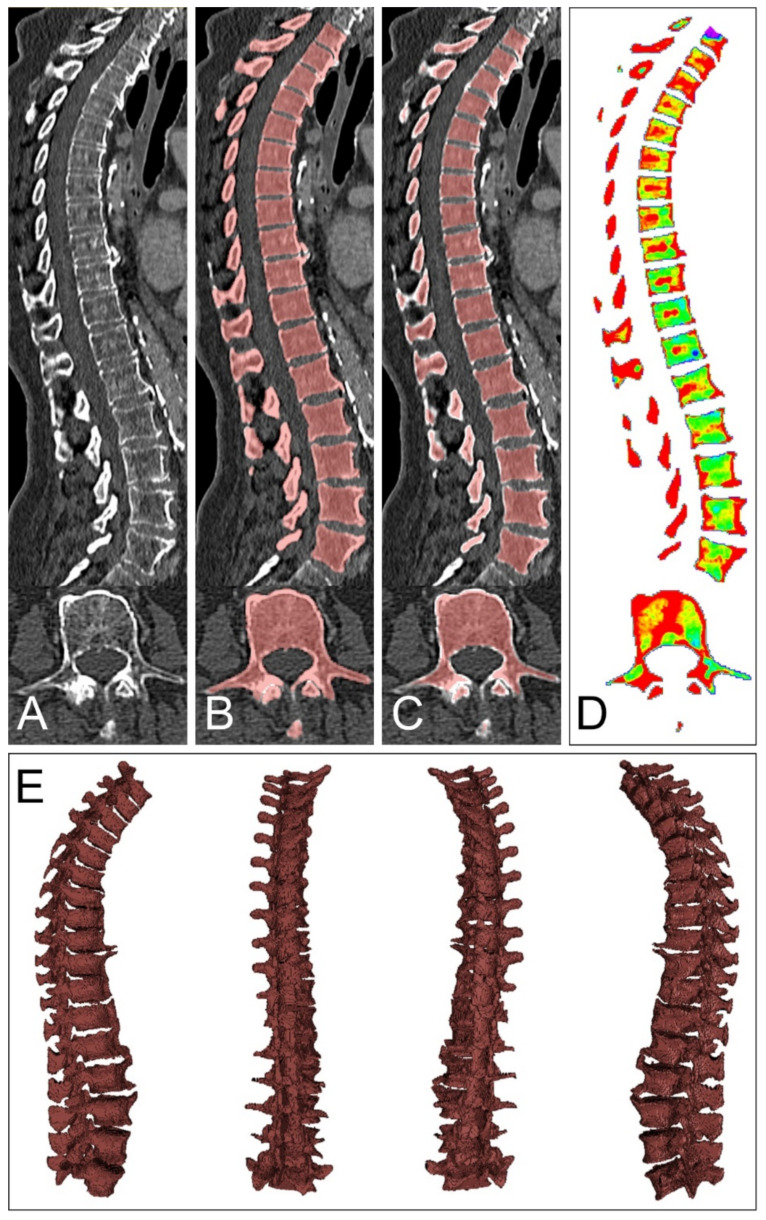
Volumetric assessment of bone marrow attenuation. Data assessment in our study was achieved by a pretrained convolutional neuronal network and did not require specific user interaction. Axial computed tomography slices served as input to the neuronal network by Payer et al. (**A**). After automated segmentation, excess vertebrae above 17, counting from the most bottom one, were excluded (**B**). Consecutively, the established volume of interest (VOI) was narrowed at each margin by 3 mm, which aimed to partially exclude the bordering cortical bone but spare the bone marrow space (**C**). Lastly, the VOI was transferred to the virtual non-calcium postprocessed data at three different calcium suppression levels ((**D**), only maximum calcium suppression shown), resulting in volumetric bone marrow attenuation data (**E**). Histograms of bone marrow attenuation were extracted for further processing.

**Figure 2 diagnostics-12-00671-f002:**
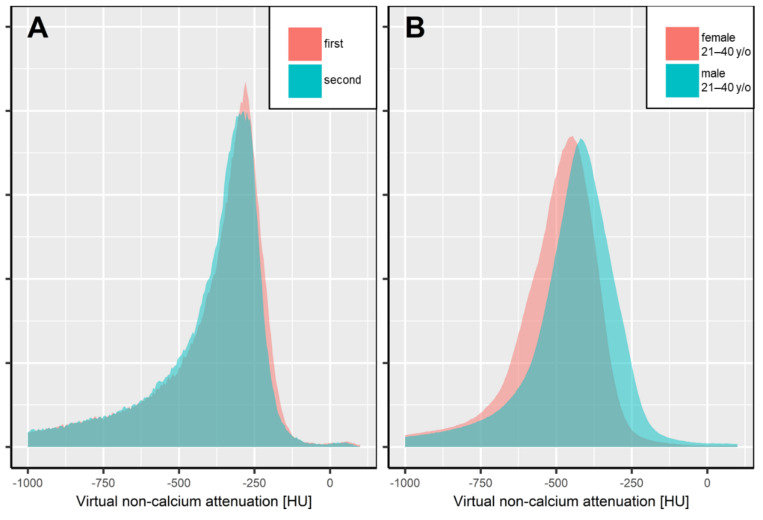
Intra- (**A**) and interindividual (**B**) overlap of automated, volumetric bone marrow attenuation assessment.

**Figure 3 diagnostics-12-00671-f003:**
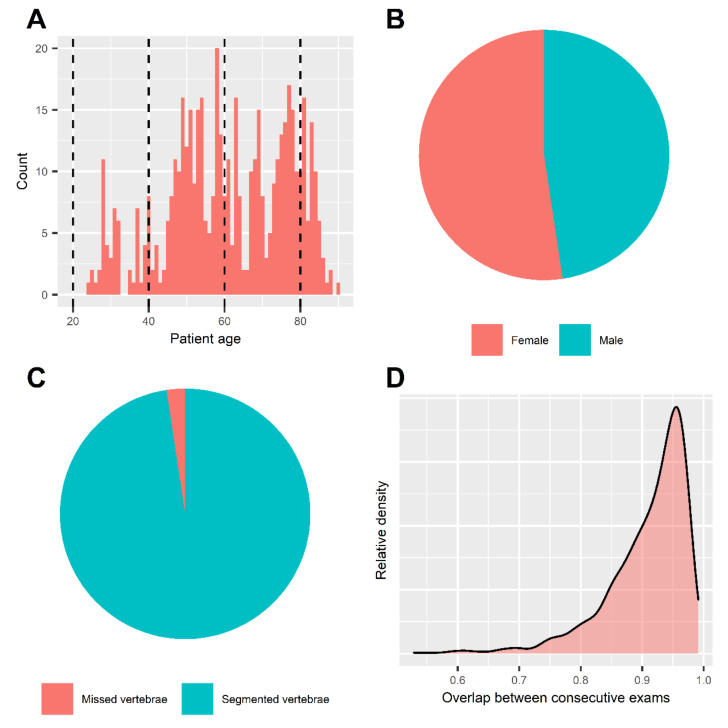
Summary statistics and patient characteristics. Histogram of patient age (**A**). Mean age of included patients was 61.0 ± 15.9 years. Dashed lines represent the borders of age subgroups for further analysis. Pie chart of patient sex (**B**). Patient sex was balanced in our dataset, including 88 females vs. 80 males. Pie chart of automated segmentation results (**C**). We achieved segmentation of 97.6% (*n* = 8298) of all included thoracolumbar vertebrae in our sample (*n* = 8500). Density chart of attenuation histogram overlap (**D**). Intraindividual consistency of our method was assessed by the overlap of attenuation histograms in consecutive exams. Median overlap was 92.9% [87.8–95.6]. 92.9% of follow-up examinations yielded at least a “good” and 65.3% an “excellent” intraindividual consistency (overlap > 80.0% and >90%, respectively).

**Figure 4 diagnostics-12-00671-f004:**
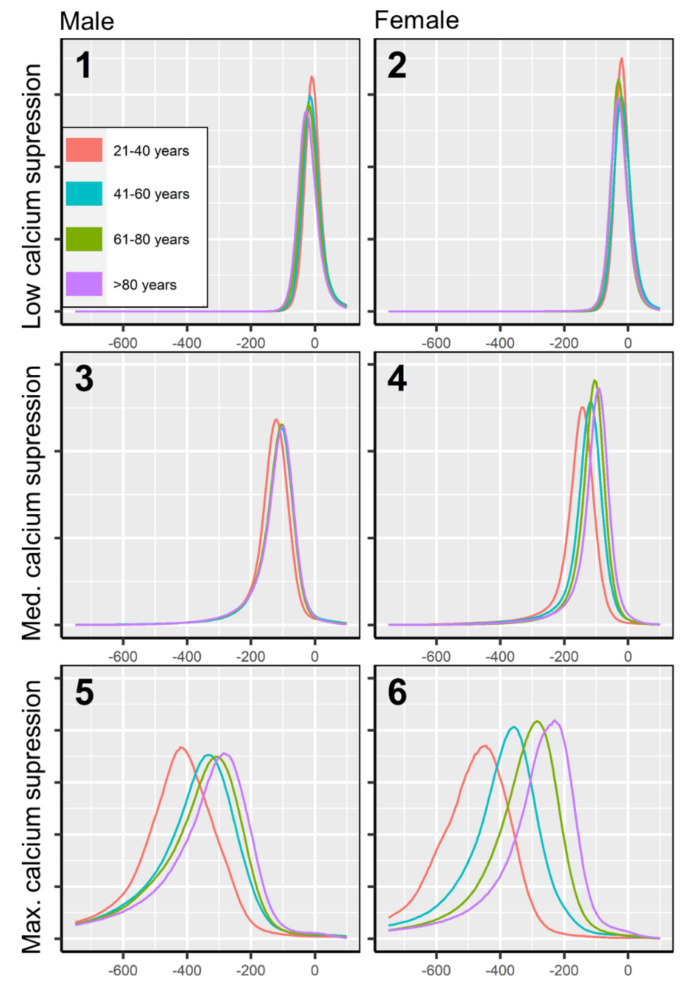
Attenuation histograms of physiological bone marrow in virtual non-calcium. Attenuation histograms are plotted for low, medium, and maximum calcium suppression (three columns), for males and females (two rows), and four age groups (four colors). The bone marrow space of each patient was normalized to a standard volume, to achieve an equal area under the curve throughout attenuation histograms for increased comparability. Bone marrow attenuation below −1000 HU was excluded from this analysis since it appears due to virtual calcium suppression of densely calcified structures (e.g., cortical bone, cortical bone islands), which do not contain bone marrow.

**Figure 5 diagnostics-12-00671-f005:**
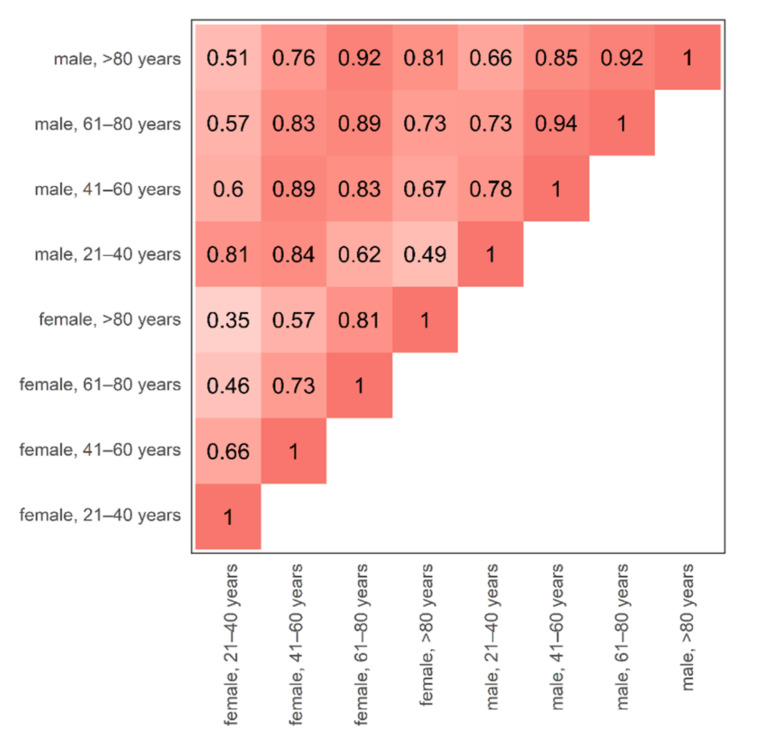
Cohort triangle of the age- and sex-specific discrepancy of bone marrow attenuation profiles. The changes in bone marrow attenuation profiles were assessed by the overlap of volume-standardized, average, virtual non-calcium attenuation histograms between all possible age- and sex-specific combinations. A high overlap of attenuation profiles is plotted by a darker shade of red. The largest discrepancy (=smallest overlap) was observed between the youngest vs. the oldest included female individuals (female, 21–40 years vs. female, >80 years, overlap 0.35). Inter-sex, age-adjusted overlap of female vs. male bone marrow attenuation was relatively high (0.81–0.89). It is worth noting that for reasons of simplicity, the bone marrow attenuation is compared at maximum calcium suppression level since this setting demonstrated the largest spread of attenuation profiles and has been described as most valuable for the diagnosis of bone marrow disease [12,23].

**Table 1 diagnostics-12-00671-t001:** Quantitative features of physiological bone marrow attenuation in virtual non-calcium (VNCa) data.

	Patient Age	Male				Female			
		Median (HU)	IQR (HU)	95th Perc. (HU)	Max. (HU)	Median (HU)	IQR (HU)	95th Perc. (HU)	Max. (HU)
Low VNCa	21–40 years	−2.5	−22.5–12.5	57.5	−7.5	−17.5	−32.5–2.5	37.5	−17.5
	41–60 years	−7.5	−27.5–12.5	57.5	−12.5	−17.5	−32.5–2.5	42.5	−17.5
	61–80 years	−12.5	−32.5–7.5	47.5	−17.5	−22.5	−42.5–−7.5	27.5	−27.5
	>80 years	−22.5	−37.5–2.5	42.5	−22.5	−27.5	−42.5–−7.5	32.5	−27.5
Medium VNCa	21–40 years	−122.5	−152.5–92.5	−47.5	−117.5	−147.5	−177.5–−117.5	−77.5	−142.5
	41–60 years	−107.5	−142.5–−77.5	−32.5	−102.5	−117.5	−147.5–−92.5	−52.5	−112.5
	61–80 years	−107.5	−142.5–−82.5	−37.5	−102.5	−107.5	−132.5–−82.5	−47.5	−102.5
	>80 years	−107.5	−142.5–−77.5	−37.5	−97.5	−97.5	−122.5–−72.5	−32.5	−87.5
Maximum VNCa	21–40 years	−422.5	−507.5–−347.5	−237.5	−417.5	−477.5	−567.5–−407.5	−312.5	−442.5
	41–60 years	−362.5	−462.5–−287.5	−182.5	−327.5	−382.5	−467.5–−317.5	−212.5	−357.5
	61–80 years	−342.5	−452.5–−272.5	−177.5	−307.5	−312.5	−397.5–−252.5	−167.5	−282.5
	>80 years	−322.5	−432.5–−247.5	−152.5	−282.5	−267.5	−352.5–−202.5	−122.5	−227.5

The physiological bone marrow attenuation is reported by median values, quartiles, 95th percentiles, and histogram maxima, grouped by patient sex, age, and calcium suppression settings (low, medium, and maximum).

## Data Availability

All data are reported comprehensively in the manuscript’s result section.

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
