# Peer review of "Dual-Energy CT, Virtual Non-Calcium Bone Marrow Imaging of the Spine: An AI-Assisted, Volumetric Evaluation of a Reference Cohort with 500 CT Scans"

_diagnostics, 2022, doi:10.3390/diagnostics12030671_

Round 1

Reviewer 1 Report

Dear Authors,

I would like to say that this is a very interesting and well-written paper that not only highlights the importance of VNCa images derived from DECT but furthermore establishes important and useful reference values that can be of use for future studies.  

Author Response

Dear Editor, dear Reviewers,

Thanks for your thorough review and for your remarks which clearly helped improving our manuscript. We followed your comments and suggestions in the revised manuscript. You will find responses for each comment and a detailed list of modifications below. Further, changes made to the manuscript are highlighted in the revised version.

We will be glad to answer further questions and looking forward to hearing your opinion on the revised version.

Sincerely,

********

Reviewer #1: Summary:
Dear Authors,

I would like to say that this is a very interesting and well-written paper that not only highlights the importance of VNCa images derived from DECT but furthermore establishes important and useful reference values that can be of use for future studies.

Thank you very much.

Reviewer 2 Report

The authors present a well-performed work on DECT virtual non-calcium bone imaging, which is a very important topic that was addressed comprehensively by the authors.

However, there are some points which I would recommend the authors to revise some points:

Title: ok

Abstract: 
"Data assessment was fully automated and did not require specific user interaction." -> Which data? This sentence seems inprecise and it is unclear to which it refers.
Otherwise ok.

Introduction
Comprehensive and adequate summary of the current status and the need for reference values.

Materials and Methods
Please give number of years of experience for the radiologists. 
Did the patient collective contain any patients with post-surgery foreign material in the spine? Were they excluded?
Probably you applied ggplot2? Or did you use a legacy version for figure creation?

Results
For Figure 3 you refer to "Basic Results" - probably a rewording would be appropriate, for example "summary statistics" or something comparable.
Text for Figure 3: " Patient sex was well balanced in our dataset, including 88 females vs. 80 males" -> Please remove the word "well"

Table 1 ist not readable easily. In the top row "(patient age)" is missing for Male patients. Please change the result presentation here (for example, additional colums for median / IQR / 95%...)
Figure 4: Here, the results are much better appreciated than in Table 1

Discussion
"In order to achieve unbiased results, we avoided specific user interaction..." -> Better: "In order to avoid bias, ..." - there may be some remaining bias due to the patient selection etc.
Otherwise ok.

Author Response

Dear Editor, dear Reviewers,

Thanks for your thorough review and for your remarks which clearly helped improving our manuscript. We followed your comments and suggestions in the revised manuscript. You will find responses for each comment and a detailed list of modifications below. Further, changes made to the manuscript are highlighted in the revised version.

We will be glad to answer further questions and looking forward to hearing your opinion on the revised version.

Sincerely,

********

Reviewer #2: Summary:
The authors present a well-performed work on DECT virtual non-calcium bone imaging, which is a very important topic that was addressed comprehensively by the authors.

However, there are some points which I would recommend the authors to revise some points:

Thank you very much.

Title: ok

Abstract:

"Data assessment was fully automated and did not require specific user interaction." -> Which data? This sentence seems inprecise and it is unclear to which it refers.

Otherwise ok.

We intended to provide the information that the entire image analysis was carried out automatically without the need for specific user interaction. We have revised the section accordingly.

Introduction

Comprehensive and adequate summary of the current status and the need for reference values.

Materials and Methods

Please give number of years of experience for the radiologists.

Revised as suggested.

Did the patient collective contain any patients with post-surgery foreign material in the spine? Were they excluded?

No.

Probably you applied ggplot2? Or did you use a legacy version for figure creation?

As you assumed, the figures were created using ggplot2, which is referred to in the methods section (l. 146).

Results

For Figure 3 you refer to "Basic Results" - probably a rewording would be appropriate, for example "summary statistics" or something comparable.

Revised as suggested.

Text for Figure 3: " Patient sex was well balanced in our dataset, including 88 females vs. 80 males" -> Please remove the word "well"

Revised as suggested.

Table 1 ist not readable easily. In the top row "(patient age)" is missing for Male patients. Please change the result presentation here (for example, additional colums for median / IQR / 95%...)

Revised as suggested.

Figure 4: Here, the results are much better appreciated than in Table 1

Thanks. As suggested, table 1 has been revised.

Discussion

"In order to achieve unbiased results, we avoided specific user interaction..." -> Better: "In order to avoid bias, ..." - there may be some remaining bias due to the patient selection etc.

Otherwise ok.

Revised as suggested.